# Coffee Yield Stability as a Factor of Food Security

**DOI:** 10.3390/foods11193036

**Published:** 2022-09-30

**Authors:** Zsuzsanna Bacsi, Mária Fekete-Farkas, Muhammad Imam Ma’ruf

**Affiliations:** 1Institute of Agricultural and Food Economics, Hungarian University of Agriculture and Life Sciences, 2100 Gödöllő, Hungary; 2Development Economics Study Program, Economic Sciences Department, Faculty of Economics, Universitas Negeri Makassar (UNM), Makassar 90221, Indonesia

**Keywords:** yield stability, climate change, coffee production, vulnerability, adaptive capability, food security

## Abstract

Yield fluctuation is a major risk in all agricultural sectors, and it influences Goal 2 (food security) of the UN SDGs. Yield fluctuations are expected due to climate change, risking stable coffee supplies, and compromising coffee-exporting countries’ ability to earn revenue to pay for food imports. Technology minimizing yield fluctuations is crucial for food security and for coffee farmers to earn a stable income. Fluctuations are small if yields remain close to the mean yield trends. In this study, the coffee yields of major producers are analyzed, together with zonal temperature data, to see where coffee is grown with stable technology under rising temperatures; thus, we demonstrate the advantages of the Yield Stability Index (YSI) over traditional stability measurements in guiding policy formulation and managerial decisions. The Yield Stability Index (YSI) is applied for 1961–1994 and 1995–2020, for the world’s 12 major coffee-producing countries. The YSI indicates that of the 12 countries, only Indonesia, Honduras, and Mexico maintain stable yield levels, while Brazil and Vietnam considerably improve their yield stability, which traditional stability measures cannot grasp. Country-wise differences exist in environmental vulnerability and adaptability, with implications for food security. The novelty is the application of the YSI, and the connection between yield stability, climate change, and food security.

## 1. Introduction

Crop production depends, to a great extent,, on uncontrollable, external conditions, e.g., weather patterns, soil, pests and diseases. Yields vary, as do production costs, profitability, and the market supply of the actual product. Yield fluctuation is a normal phenomenon, considering annually varied weather and environmental conditions. However, good production technology can mitigate the adverse effects caused by typical environmental anomalies; therefore, yield fluctuations should remain within a reasonable range of the average yield trend. To develop production technologies and crop varieties that can tolerate adverse environmental effects and produce high yields. it is important to respond to climate change and maintain environmentally and economically sustainable production.

The yield stability of coffee is closely related to several of the 17 Sustainable Development Goals of the UN [1]. Food security is defined as SDG2 (end hunger, achieve a secure food supply and improved nutrition, and promote sustainable agriculture). The research on climate-induced changes in food availability and agricultural land use indicate considerable overall losses of cropland and forests, often with different predicted patterns for the Northern and Southern Hemispheres [2,3] in relation to forecasted population growth and agricultural technology improvements [4,5]. Possible responses have also been forecasted both globally and regionally [6,7,8]. For coffee-producing countries, SDG2 is closely related to SDG1 (end poverty in all its forms everywhere), SDG8 (promote sustained, inclusive, and sustainable economic growth, full and productive employment, and decent work for all), SDG10 (reduce inequality within and among countries), and SDG13 (take urgent action to combat climate change and its impacts), considering that most coffee producers are small-scale farmers whose main income source is selling coffee beans, and their farms are not diversified enough to produce food staples for their own consumption [9]. So, coffee production is a means of income generation for these farmers, contributing to job creation for family members and in the primary processing of green coffee beans before export.

Literature on food security analyses has, as its primary focus, only one dimension, namely agricultural output as a proxy for food availability. However, food security also comprises the availability, access, utilization, and stability dimensions [10]; however, assessment of the stability aspect of food security is very uncommon, involving some dynamic measure of outcomes over a given time horizon, measuring the deviation of the time series from a given threshold [10].

Food security refers to a situation in which all people, at all times, have physical, social, and economic access to sufficient, safe, and nutritious food for an active and healthy life. This definition includes the concepts of food availability, food access, and how food is utilized. Small-scale coffee producers are trying to maintain a sustainable livelihood with modest land holdings, high levels of initial capital investments in their coffee plants, and a vulnerability to a volatile international price structure for their cash crop; in addition, they live in countries with relatively weak trade positions. With higher coffee prices in the international market, the desire for a profitable cash crop often encourages farmers in traditional coffee-growing areas to increase their production, leaving fewer and fewer areas for food subsistence. Export crops, such as coffee, may offer the promise of a better life, and an escape from the poverty trap of subsistence agriculture. Wishing to participate in a cash economy, many smallholder coffee farmers allocate their investments to coffee and away from subsistence food production, hoping that the extra money will allow for additional food purchases. The isolated rural areas where the world’s best coffee is grown are exposed to multiple food insecurity risk factors: the depletion of natural resources, environmental degradation, shocks such as natural disasters, and seasonal changes in food production and food prices. Coffee farmers face the instability of green bean coffee prices and of fluctuations in food prices, and this increases the food vulnerability of these communities [11].

Food safety is linked to food security through health and livelihoods. The presence of food hazards can lead to food losses and reduced food availability for food-insecure populations. Food safety also increasingly plays a role in producer livelihoods, as smallholders seek to meet requirements in high-value markets, particularly exports. As consumers often lack information about food safety hazards in specific food products, producers cannot obtain rewards for supplying safer food; therefore, they may not have sufficient financial resources for the extra costs of food safety. Sometimes producers may also have little idea of the existing hazards in their products, so they cannot react and improve food safety [12]. In developing countries, efforts to improve food safety have been particularly focused on exports to high-income countries. Compliance with food safety standards in high-income countries demonstrates the costs of such improvements and the way in which compliance leads to higher incomes for developing country smallholders, thus supporting improved food security [12]. To satisfy demands for food safety, retailers and manufacturers increasingly use sustainability-oriented standards and labels, especially for luxury foods such as coffee, tea, cocoa, or tropical fruits. For coffee, the global market share of products with sustainability certification has been growing since 2006 [9,13]. Certified coffees are assumed to have a higher price, but in reality, effective price advantages are often minimal for farmers. As data from coffee farmers in Mexico and Peru show, yields may be more important than price premiums for increasing net returns. There are considerable regional differences in average yield and quality levels that influence the incomes and livelihoods of coffee farmers, influencing both the food safety and food security of coffee-growing rural regions [13].

Although climate change and food supplies have been extensively studied in the literature, they have not been analyzed specifically from a yield stability viewpoint, which is a novel aspect of the present research. The analysis applies the Yield Stability Index developed in 2002 [14] to assess the yield fluctuations in major coffee-producing countries of the world.

Coffee is commercially produced in more than 50 countries, and it is one of the most-traded agricultural commodities in the world. There are 12.5 million coffee farms worldwide; about 95% of these are smaller than 5 hectares and considered ‘smallholders’, with at least 5.5 million of them living below the international poverty line of USD 3.20 a day. Smallholder producers are predominantly located in 20 countries where the climate and soil are suitable for growing coffee, including Ethiopia (2.2 million), Uganda (1.8 million), and Indonesia (1.3 million), Vietnam, and Colombia (each having more than 500,000 farmers). The producing countries are situated geographically between the tropics, and they are typically low- to lower-middle-income countries, capturing the lowest value along the value chain [9]. The coffee itself represents just a small percentage of the value of the final product. Two main coffee species are produced, traded, and consumed. Arabica, grown at higher altitudes, makes the world’s finest-tasting coffees, and it accounted for 55%–60% of the world’s supply from 2011 to 2017. Robusta, which is grown at lower altitudes and is hardier, though yielding a lower quality, could play a key role in meeting the challenge caused by climate change [9].

In recent years, the price of green coffee has been unstable and generally very low, e.g., only 2.71 USD per kg in April 2021, which was hardly sufficient to cover production costs, making the future for many coffee farmers in the tropics precarious. However, from summer of 2021, prices started to speedily increase, reaching 3.54 USD per kg in August 2021, 4.51 USD/kg in January 2022, and 4.21 USD/kg in July and August 2022 [15]. This is partly due to the international trading on the stock exchange and partly due to weather and climate conditions influencing yields. Coffee consumption did not considerably change during the COVID-19 pandemic situation in Europe, America, or Asia, in spite of the confinement that restricted consumption to homes and other closed environments [16]. Climate change and higher temperatures are threatening the production of premium coffee, especially in Latin America, while climate variations and altitude considerably influence coffee quality [17,18,19]. Coffee grows optimally between 800 and 1400 m above sea level; however, due to global warming, this elevation is forecasted to increase to 1200–1600 m by 2050, assuming a 2.5 °C rise in temperature, and accounting for 0.65 °C cooling per 100 m elevation [20]. The adaptation process means that if coffee farmers do not raise the elevation of their fields, the productivity and quality of their coffee will decrease. Additionally, rising temperature and increased water stress also affect the migration of pests (e.g., moths and grasshoppers) and may prolong their flying period, allowing their spread over longer distances, as well as the explosion of pest populations due to increased breeding rates. Add to this the decrease in suitable growing areas and declining coffee prices, and it is clear that food safety, food security, and the welfare and livelihoods of farmers, especially smallholders, are under threat [9,21]. Although coffee has played a relatively small role in global deforestation so far, if climate change gradually drives production into new areas, this may threaten the last intact primary forests on our planet [20,22,23].

The major countries producing green coffee beans are presented in Table 1, ranked according to their share in global production. These 12 countries covered 88.36% of the global output and 83.1% of the total export in 2020 [24].

The value of coffee exports represents a rather high proportion of the total agricultural exports, ranging from 1% in Mexico to 47% in Ethiopia, and its share is also quite high in the total merchandise export of these countries [24]. As coffee is a core component of the export revenues for most of these producer countries, it has an essential role in covering the import payments (Table 2). The analyzed countries are seriously dependent on food imports for domestic supply, and are less than self-sufficient in this respect (Figure 1). Coffee export revenues can cover about half of the food import expenditure in Brazil, Colombia, Ethiopia, Honduras, Nicaragua, and Uganda. The situation for Ethiopia, Honduras, Nicaragua, Colombia, and Uganda seems to be rather serious, because in a bad coffee year, they do not have many other exportable products to cover their food import needs, and coffee farms are not diversified enough to produce a significant amount of stable food [25]. Brazil is in a somewhat better situation as coffee brings only 2.7% of its total export revenue, which might be substituted with something else if needed.

The prevalence of severe food insecurity and of undernourishment in the population are quite high in some of these countries [26], having increased since 2016 and reaching around 20% of the population in many countries, with only a few exceptions (see Table 3).

Coffee exports represent a substantial proportion of food imports for these countries, often reaching 40–50% of the share. This means that these major coffee-producing countries—except for Vietnam, Indonesia, and to some extent, Mexico—really depend on food imports. Therefore, insecure coffee yields resulting in insecure coffee export revenue can be devastating for their welfare. A clear picture of coffee yield stability is crucial to making the necessary technology improvements and developing adaptation strategies as soon as possible.

Yield fluctuations generate fluctuations in farmers’ sales revenues as well as in the supply of particular products [27,28]. Although agricultural support policies may mitigate the farmers’ risks associated with yield fluctuations [29,30], the development of stable technologies can respond to the changing environmental conditions and allow high-but-stable yields of major importance. Various measures have been used to evaluate yield stability [31]: the sum of the percentage or squared difference between actual and baseline yields over time for wheat yields in Australia [32] and rice in Nepal [33]; the squared sums of the differences between actual annual yields and average yields for maize and legumes in South Africa [34] and oats in the UK [35]; yield standard deviations for rice in Bangladesh [36]; Singular Spectrum Analysis (SSA) applied in a South-Korean study [37]; and time series fluctuations around a linear or non-linear trend measured using statistical dispersion indicators, e.g., the sum of the absolute errors (the absolute difference between actual values and trend estimations), the standard deviation or the coefficient of variation [38,39,40], or the average percentage deviation from the trend [41,42].

Empirical studies support the fact that high yields usually imply higher yield variability, which is often better-tolerated by producers than lower yields with lower yield variance. Therefore, when measuring yield stability, variance levels should be assessed together with mean yield levels, focusing on the risk of the yield falling below a certain limit [38,43,44]. When assessing crop yields from an economic viewpoint, the focus is on finding varieties that turn out relatively stable yields under varied environmental conditions typical for a country or a larger region of production. Such analyses have been conducted for wheat and maize [45], barley [44], maize [46], and winter wheat [47] (by applying the coefficient of variation, standard deviation, and variance around the trend line for measuring variability) and for paddy rice, maize, wheat, and rapeseed [48] (by measuring the proportion of residuals and the estimated trend values, with negative residuals considered a problem).

The problem with standard deviation, the coefficient of variation, and residuals around the trend is that a high standard deviation (or CV or sum of residuals) may be due not only to a few very extreme fluctuations, but also to many small ones. However, many small fluctuations are acceptable and tolerable, while the occasional occurrence of extremely low or high yields may create serious economic risk [14]. Therefore, the Yield Stability Index should be able to distinguish many small fluctuations from a few extreme ones. The Yield Stability Index (YSI), developed by Bacsi and Vízvári in 2002 and later improved by Bacsi and Hollósy [14,49] in 2019, is such a measure, tested for 10 countries and 18 major crops produced in Europe in 1961–2000 and 2004–2016. These studies compared the yield reliability of crops, reflecting production technology and identifying the suitability of crops and technologies to the external conditions of the area. The term “weakly technologized crop” was introduced for crops with large yield variability in a given time period.

The objective of the present research is to compare major coffee producers to see where coffee is grown with stable technology under rising temperatures, and to demonstrate the advantages of the Yield Stability Index (YSI) over traditional stability measurements in guiding policy formulation and managerial decisions.

## 2. Materials and Methods

*Data:* The FAOSTAT database [50] was used to select coffee (green beans)-producing countries. Yield time series from 1961–2020 were retrieved from this database and used for the analysis.

*The choice of time period:* The FAOSTAT data are available from 1961 to 2020, and the whole time span was chosen for the analysis. However, considering environmental changes during the past several decades, the time period was broken up into two segments, according to climate trends. The Goddard Institute of Space Studies is one of the major institutions dealing with climate change research. They publish annual temperature change data for the major geographic zones of Earth [51]. These time series were used for segmenting the 60 years of analysis. Figure 2 presents the average temperature changes globally, and separately for the Northern and the Southern Hemispheres, showing an obvious increase in all three time series. The increase is not linear; its rate obviously increases with time. A quadratic trend was fitted to each of the three time series; the equations and R^2^ values are shown in the figure. The trend equations (x denotes the year 1960, i.e., 1961 is x = 1) are:

Temp(Glob) = 0.000143x^2^ + 0.008748x − 0.08414, R^2^ = 0.9168

Temp(N-hemisphere) = 0.000362x^2^ + 0.000216x − 0.04971, R^2^ = 0.9038

Temp(S-hemisphere) = −0.0000703x^2^ + 0.01696x − 0.11442, R^2^ = 0.8739

The choice of quadratic trend was motivated by the changing patterns of the Northern and Southern Hemispheres. The GISS temperature series in [51] show that the global temperature anomalies follow a nonlinear trend, and lowest degree polynomial that fits this trend is the quadratic one. As Figure 2 shows, at the beginning of the analyzed period, the temperature change patterns are approximately flat in both hemispheres, while by the end, the trends are quite steep; thus, a nonlinear trend line is more reasonable than a linear one. However, higher-order polynomials did not give a reasonably improved fit, and the coefficients for the higher-order factors were all very small (a degree of 10^−6^ or smaller). Thus the quadratic trend equations were used for the analysis. Simple algebra shows that the trend lines of the Northern Hemisphere and the Southern Hemisphere meet at 1994 (x = 34, year = 34 + 1960 = 1994). Before 1994, the three trends move very close to each other, while from 1995, the three curves tend to move away from each other, with the Northern Hemisphere warming up more quickly than the Southern Hemisphere. Therefore, the breakpoint is 1995, i.e., the first time period is 1961–1994 (34 years), and the second one is 1995–2020 (26 years).

Country-wise temperature anomaly time-series were not available for the present research, but there are time series for the various latitude ranges relevant to coffee-producing regions, i.e., the 24° N–24° S tropical zone, and for the 44° N–24° N and 44° S–24° S zones. The ’normalized’ temperature change data (i.e., values divided by the series 60-year average) show increasing linear-like trends, with the tropical zone having the highest values at the beginning, but the lowest ones by the end of the analyzed period, with the smallest linear slope. Meanwhile, the 44° N–24° N zone, starting from the lowest level and ending at the highest level, has the steepest slope.

The relevant linear trend lines are: for 44° N–24° N: y = 0.058135x − 114.767433; for 24° N–24° S: y = 0.041567x − 81.735987; and for 44° S–24° S: y = 0.047446x − 93.468495 (see Figure 2b and Table 4). Comparing the three trend lines, they meet approximately at 1994–1995. The 10-year moving averages for the three relevant zones (24° N–44° N, 24° S–24° N, and 24° S–44° S) also show a breakpoint at 1995; up to 1995, the highest moving average is the 24° S–24° N region, and the smallest one is the 24° N–44° N zone, but from 1996, the highest value is 24° N–44° N (the former coolest region), while the lowest value is 24° S–44° S, and the 24° N–24° S region is between them. The 1990–1999 section of the time series is illustrated in Figure 2c and Table 4. This supports the idea that it is meaningful to use the years 1994–1995 as the break-point for our analysis.

*Countries chosen for the analysis:* The countries were selected according to their share of global production. As the breakpoint in time was 1995, the years 1995 and 2020 were referred to, and the countries with at least 1% of total global production in 1995 and 2020 were chosen (Table 1). Altogether, this included 12 countries covering 88.36% of global output and 83.1% of total global coffee green bean exports in 2020. Therefore, the rest of the analysis deals with the following countries: 3 countries in South America (Brazil, Colombia, and Peru), 4 countries in North America (Guatemala, Honduras, Mexico, and Nicaragua), 3 countries in Asia (India, Indonesia, and Vietnam), and 2 countries in Africa (Ethiopia and Uganda).

*Computation of the Yield stability Index (YSI):* The statistical evaluation of yield stability was conducted through the Yield Stability Index by Bacsi and Vízvári in its revised form by Bacsi and Hollósy [14]. A detailed description of the computation of this index is given in [14]. A brief overview of the calculation of this index is given below.

Taking a country and a particular crop, annual yields are measured for a given period of years. As the magnitude of the fluctuations naturally depends on the magnitude of the whole time series, the first step of the computation is to express the yield values in proportion to the overall average of the whole time series; therefore, the extent of the fluctuations is assessed compared to the mean yield. This makes the index values comparable for various crops and various countries, even if their average yield levels differ due to the intensity of production or to climatic conditions; however, it does not distort the relative extent of the fluctuations. Then, a linear regression line is fitted to the rescaled yield series, representing the mean trend of yield growth. The residuals (the differences in the observed rescaled yields from the linear trend values) are computed, and these residuals represent the actual yield fluctuations. The magnitude of these residuals is evaluated by comparing them to a normal distribution of the same variance as the residual series (when several countries are compared, the variance is the average of the residual variances of all countries). Taking the range of the residual values, this range is divided into deciles, i.e., 10 equal segments (again, when more than one country is analyzed, the range is defined by all the countries’ minimum and maximum residual values). Then, the distribution of the actual rescaled residual values is compared to the normal distribution of the zero mean and the variance of the residual series. The fluctuations of the yield series are considered high when ‘many’ residual values are far from zero, i.e., when many residual values fall to the lowest three deciles or the highest three deciles. The residuals falling into one of the four central deciles (those close to zero) are considered small fluctuations. The term ‘many’ is understood in comparison to the above-defined normal distribution. The proportion of the residuals falling into the central four deciles and the proportion of such values derived from the defined normal distribution are computed, and the difference between the observed proportion and the normal proportion is computed. A positive difference is favorable; a negative difference is unfavorable. Similarly, the proportions of the residual values falling into the 6 segments far from zero are computed, and then, the same proportion as that derived from the normal distribution is deducted from this. This value is the unfavorable difference. Stability requires a large favorable difference and a small unfavorable difference. Then, the YSI is computed by deducting the unfavorable difference from the favorable difference. The theoretical value of YSI falls between −2 and 2 (as the values of the involved proportions fall between 0 and 1). The computation method gives an index that is directly comparable for time periods of different lengths and crops with different magnitudes of mean yields. For more technical descriptions, see [14].

The present paper uses this methodology to compute the YSI for the coffee yields of 12 countries between 1961 and 2020. To illustrate the benefits of this index, the coefficients of variation (CV% = 100 × standard deviation/average of the series, %) are also computed for the same data.

## 3. Results

### 3.1. Yield Time Series, 1961–2020

The time series of yields show considerable volatility during the analyzed period (Figure 3). As Figure 3 shows, during 1961–1985, yield data fluctuate around 500 kg/ha, varying between a 160 and 1000 kg annual yield per hectare (1961–1985), but then, the data series become increasingly unstable. The Vietnam yields show a sharp yield increase up to 2700 kg/ha, Brazil up to 1950 kg/ha, while Mexico shows a slightly decreasing pattern ending up at around 400 kg/ha, and most other countries have yields between 500 and 1200 kg/ha.

The average yields and CV% values are given In Table 4. As the CV% (standard deviation divided by the series average, in %) show, the countries differ considerably, with Indonesia having quite uniform yield values, with only 7.3% CV; the other extreme is Vietnam, with 62.9%, and Brazil, with 51.1%. As these two countries are the largest producers in the world, their unstable yields may have serious consequences for global supplies and global market prices. The residual series were tested for normality using the Shapiro–Wilk test. The *p*-values of the test are shown in the last column of Table 5. As these values indicate, all the residual series are of normal distribution at the 0.05 or 0.01 level; therefore, the logic of the YSI computation comparing the residuals to a normal distribution is meaningful.

The fourth column of Table 5 shows the YSI values for 1961–2020. Negative values are measured for Brazil, Vietnam, Ethiopia, India, Uganda, and Nicaragua, meaning weak technological stability. The weakest is Ethiopia, with the smallest YSI value, followed by Brazil, then Uganda, India, Nicaragua, and Vietnam.

Comparing the information gained from the CV% and the YSI, a smaller (negative) YSI means more instability, while a higher CV% means more instability. Therefore, if the YSI did not provide any additional information to that provided by the CV%, a negative linear relationship was expected. However, as we see in Figure 4, this negative trend does not prevail, the best fitting linear trend shows only R^2^ = 0.0401 for the goodness-of-fit coefficient, and Brazil, Ethiopia, Honduras, Indonesia, and Vietnam are particularly far from the fitted line. As Figure 4 shows, Ethiopia has a lower CV% than Honduras, indicating smaller volatility, while the negative YSI for Ethiopia points to greater instability than the rather stable Honduras. The same is true when comparing Brazil with Vietnam, or Uganda with Nicaragua. However, when comparing Nicaragua with Colombia, both the CV% and the YSI show Colombia as more stable than Nicaragua. Therefore, the information content of the CV% is different from the YSI.

As we indicated earlier, the 60-year time period is not homogeneous; the temperature trends show a considerable speed-up from around 1995. Therefore, it is reasonable to analyze the two time periods (1961–1994 and 1995–2020) separately.

### 3.2. Separate Analysis of the Time Periods 1961–1994 and 1995–2020 

As the negative values of YSI in the fourth column in Table 6 show, considerable yield fluctuations characterize half of the analyzed countries in the first 34 years. Vietnam, Brazil, and Uganda are especially unstable, but Nicaragua, India, and, to a lesser extent, Ethiopia are in this category.

The seventh column of Table 6 shows YSI values for the last 24 years. The unstable countries are Vietnam, Colombia, Ethiopia, Uganda, and Nicaragua. The last column of the same table shows the change in YSI; positive values mean an improvement in stability, and negative values mean a deterioration. India and Brazil could stabilize their coffee yields in spite of increased temperature rises, while Nicaragua and Ethiopia have become even more unstable. Vietnam and, to a lesser extent, Uganda, though still a little unstable, considerably improved their stability situation, while Colombia changed its formerly stable position to a slightly unstable one. The CV% values show a different picture, as is illustrated by the third and sixth columns of Table 6.

During 1961–1994, the countries with the largest CV% are Vietnam, Honduras, Nicaragua, and Brazil, while the lowest values are for Ethiopia (only with 2 years of data), Indonesia, and Peru (all under 10%). Vietnam (large volatility) and Indonesia and Peru (small volatility) are evaluated according to both CV% and YSI in the same way. Honduras and Guatemala, on the other hand, are valued positively according to the YSI but rather badly according to the CV%, and Uganda and Brazil are assessed similarly to Columbia and Guatemala according to the CV%, while according to the YSI, the former two countries are much worse than the latter ones (Figure 5).

During the 26 years from 1995 to 2020 the yield time series seem to fluctuating less than in the former period. The yields of Vietnam and of Brazil are considerably higher than those of the other 10 countries, but visual analysis (Figure 6) does not suggest wider fluctuations. Table 6 gives the overall average yields and CV% values for the whole period. This also suggests that yields fluctuate less in spite of the increasing temperature trends, which usually imply more changeable weather and more frequent extreme weather events.

As the negative YSI values indicate, in 1961–1994, Uganda, Nicaragua, Vietnam, and Ethiopia were weakly technologized, as were Brazil and India. In 1995–2020 Nicaragua, Uganda, Vietnam, and Ethiopia were still weakly technologized, but Vietnam and Uganda improved a little, while Brazil and India moved out of the weakly technologized position, and Colombia deteriorated and became weakly technologized. The highest YSI values are achieved by Mexico and India, followed by Indonesia, Honduras, then Brazil in 1995–2020, while in 1961–1994, the highest values were for Honduras, Guatemala, Peru, Indonesia, Colombia, and Mexico; however, these YSI values are all smaller than those of the best five countries in the second period (Figure 6).

The comparison of YSI and CV% values shows differences in this time period, too. Brazil, with the highest CV% (large fluctuations), has a positive YSI (rather stable yields), and the same is true for Mexico—i.e., the YSI evaluates these countries to be quite contrary to the CV%. Mexico and Nicaragua have similar CV% values, but Nicaragua has weakly technologized yields according to the YSI, and Mexico is judged to be stable according to the YSI. Vietnam, Honduras, and Guatemala are evaluated similarly according to the CV% and YSI; Colombia and Ethiopia are unstable, with negative YSI values, but their CV% is modest, indicating a stable yield trend (Figure 6).

### 3.3. Comparison of Yield Volatilities during 1961–1994 and 1995–2020

Table 5 and Figure 6 show that the countries’ performance differs in terms of yield stability: Brazil and India improved the most (both from a weakly technologized to a well-technologized state), followed by Vietnam (still weakly technologized, but much closer to the well-technologized status). Mexico, Indonesia, and Honduras improved their former well-technologized status to an even higher YSI. Peru and Guatemala decreased their YSI but still remained positive, and Uganda improved a little but was still negative. At the same time, Colombia, Ethiopia, and Nicaragua worsened their technology levels, ending with negative YSI values.

Figure 6 compares the YSI values for the two analyzed time periods. The green markers show the countries with decreasing fluctuation, i.e., improved yield stability, while the red markers show those with deteriorating yield stability. The 1:1 line outlines the no-change situation.

As is presented, 7 of the 12 countries were able to improve their yield stability values, and of these, Brazil and India actually improved their YSI values from negative to positive, in spite of the warmer temperatures of the second period. The improvement in Vietnam and Uganda is also considerable, though their yields were still somewhat unstable in the latter time period. Mexico and Indonesia also produced more stable yields than in the first period, though their yields were relatively stable at that time, too.

Nicaragua and Colombia considerably worsened their positions (with Colombia actually moving from positive to negative), while a smaller decrease was experienced by Peru, Guatemala, and Ethiopia.

Figure 7 presents the YSI values against average yield values during the two analyzed periods. The general tendency is an average yield increase with increasing YSI values, which is a beneficial tendency for global supply, although some countries achieved higher stability with lower yield averages. The position of Vietnam is outstanding, with a 300% average yield increase and a considerable improvement in stability, but Brazil, India, and Honduras are also remarkably improved.

## 4. Discussion

Yield levels and yield stability are equally important for farmers’ livelihoods (profitability) and for market supply. The optimal cropping technology should provide high yields and low variability for long time periods under varied weather and environmental conditions.

The present analysis revealed that major coffee-exporting countries produce yields that are stable to different extents. The Yield Stability Index (YSI), applied to the time periods 1961–1994 and 1995–2020, revealed that in spite of the second period being characterized by more markedly rising temperatures, the largest producers, i.e., Brazil and Vietnam, could considerably improve their average yield levels, together with their yield stability. Similar tendencies, though to a smaller extent, were found for India, Indonesia, and Honduras, though the yields in Indonesia slightly decreased (to 94% of the first time period). This means that these five countries yield green coffee beans using technologies that seem to be able to handle adverse weather effects and other external risk factors reasonably well; therefore, their coffee production can be considered sustainable in the long run.

Uganda showed a very small yield increase and slightly improved yield stability, basically performing at the same level in the second period as in the first one. Ethiopia is very similar, but the slight changes are experienced in the opposite direction: a small yield decrease and small yield stability deterioration. As both countries have rather low yields and negative yield stability values, their production technologies do not seem good enough to be sustainable in the long run.

There are countries for which increasing yields go together with increasing yield fluctuations, i.e., deteriorating yield stability. Nicaragua, Colombia, Guatemala, and Peru are similar in this respect. Still, while the Yield Stability Index in the first two countries is negative during 1995–2020, the YSI values are positive for the former two countries, although worse (smaller) than in the first period. This shows that for these four countries, the production technology can lead to high yields in good years, but at high risk, which makes the livelihoods of small-scale coffee farmers very uncertain in the future.

Mexico is a special case because its average yield decreased considerably from 1961–1994 to 1995–2020, to 69% of the former level, but its yield stability improved a great deal. With this performance, Mexico has the lowest average yield among the 12 countries in the second time period, and only two other countries with decreasing yields still produce more than 90% of their former yields (Indonesia: 94% and Ethiopia: 96%). Mexico, with such a yield decrease, reaches the level of only two-thirds of the second lowest average yield (Indonesia) and is highly unlikely to remain competitive in the long run.

Thus, among the producers of both the northern and the Southern Hemispheres, there are some who cope well with the rising temperatures and changing environment, and this indicates that countries may have suitable technological responses to mitigate adverse conditions. Yield variability, of course, is not entirely determined by temperature, and many other factors may be responsible for yield fluctuations, i.e., socioeconomic conditions or agricultural policy-related issues, which were not included in the present research. Yield stability results may have an implication for the agricultural support policies of countries. These may be associated with technology improvements that can lead to sustainable production in the long run or with short-term mitigation, such as general financial support or insurance schemes for protection against risk; however, these may decrease the sector’s motivation to adopt production technologies more appropriate for the changing environment.

As Brazil, Vietnam, Honduras, India, and Indonesia seem to be the countries coping well, they are examples for the Northern and the Southern Hemispheres to handle the challenges raised by climate change. Of course, other factors, including economic and political crises, may also negatively affect the agricultural production of countries, and it may easily explain the bad performance of some Latin American countries. This is in line with research discussing productivity gains due to technological improvements in Brazil, [52] including increases in plants per hectare, the use of agricultural machinery, the development of new varieties, and the adoption of irrigation techniques, while still exhibiting seasonal variation in productivity.

Uganda, Guatemala, and Ethiopia are the most food-insecure countries (no data are available for Nicaragua in this respect), and Ethiopia, Guatemala, Peru, and Nicaragua have the highest rates of undernourished population (no data are available for Uganda). These five countries also considerably depend on coffee export revenue to cover their food import expenses (16–61% in 2020); therefore, unsustainable and risky coffee yields can have detrimental consequences on their food supply on a national level while leading to poverty for the smallholder coffee farmers, too. Coffee area suitability, assessed via climatic and soil requirements, indicated that climatic factors are the most limiting factor, and negative climate impacts dominate in all the main producing regions [22,23,53]. However, the actual expected yield was not considered; the forecasts were based on the expected changes in the environment. Former research for Indonesia and Ethiopia also supports our findings. Yield and production differences were found to be due to extreme climatic conditions and relatively low coffee yields due to weather and to technology inefficiencies in Indonesia [54,55]. Yield trends for Ethiopia between 1993 and 2019 showed a continuous decrease due to inefficient technology, which highlights the need for improved varieties and agronomic practices to be changed [56]. Our analysis, though not directly targeting the impacts of climate change, shows that countries have adapted to different extents to the changing climate so far. Such yield stability evaluations for coffee do not exist in the literature, to our best knowledge.

The present analysis also revealed, through the example of the major coffee producers, that the YSI is a more suitable measure of yield stability than the traditional indicators based on various measures of standard deviation. While the CV% values pointed to Brazil, Mexico, Uganda, and Nicaragua as the most unstable countries, Indonesia, India, Ethiopia, Peru, and Guatemala were shown as the most stable ones. These results are rather contrary to the YSI-based assessment, and underline the importance of distinguishing between the occurrence of frequent small fluctuations and of rare but large ones. These findings are in line with the results of Bacsi and Hollósy [14,49].

## 5. Conclusions

The potential policy implications of the present research are two-fold. First, it can reveal differences among coffee-producing countries and technologies, and identify which countries have been capable of improving their technology in response to changing environmental conditions. On the other hand, the results also reflect the differences in the environmental threats, and point out the most vulnerable production areas. These can direct the need for intervention in production and support programs. The results, therefore, can be used to guide future international and national support programs as well as agricultural extension programs, both from a food-security viewpoint and for the issues of welfare, income, and job security related to the coffee-producing sector.

The relationship between the YSI and climate variables is an exciting issue. However, the YSI is one value for a series of years; therefore we have only three values for each country: one for 60 years, one for 34 years, and one for 26 years. Computing correlations with three values is not very meaningful. A future research topic could be to compare YSI values for shorter time segments—e.g., 10-year periods—and compared to the 10-year mean temperature changes to see how YSI values are influenced by changing temperatures. Further research may be carried out to reveal the relationship between annual coffee yields and annual mean temperatures or other weather indicators in producing regions, with regard to the stability of production.

Another area of further investigation is the yield stability differences between the two major coffee varieties, Arabica and Robusta. Unfortunately, publicly available databases for the producer countries do not provide yield data by coffee variety; therefore, more sophisticated data collection methods are needed for this analysis.

## Figures and Tables

**Figure 1 foods-11-03036-f001:**
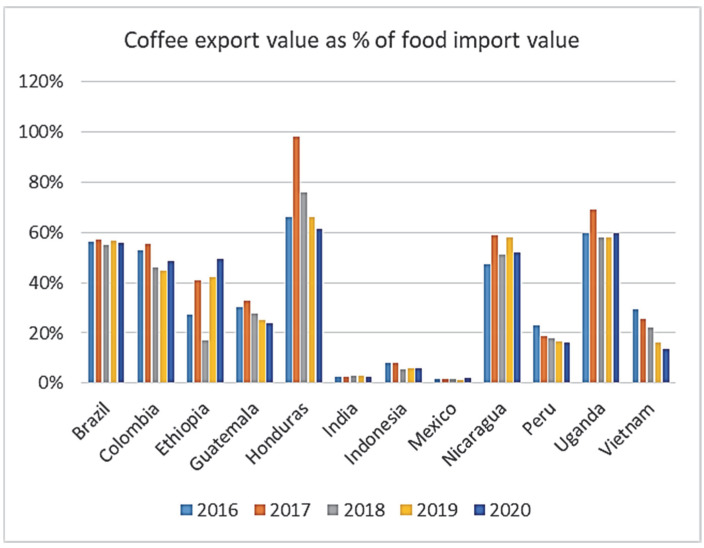
Coffee export revenues as % of food import expenditure, 2016–2020. Source: authors’ elaboration based on data from [24].

**Figure 2 foods-11-03036-f002:**
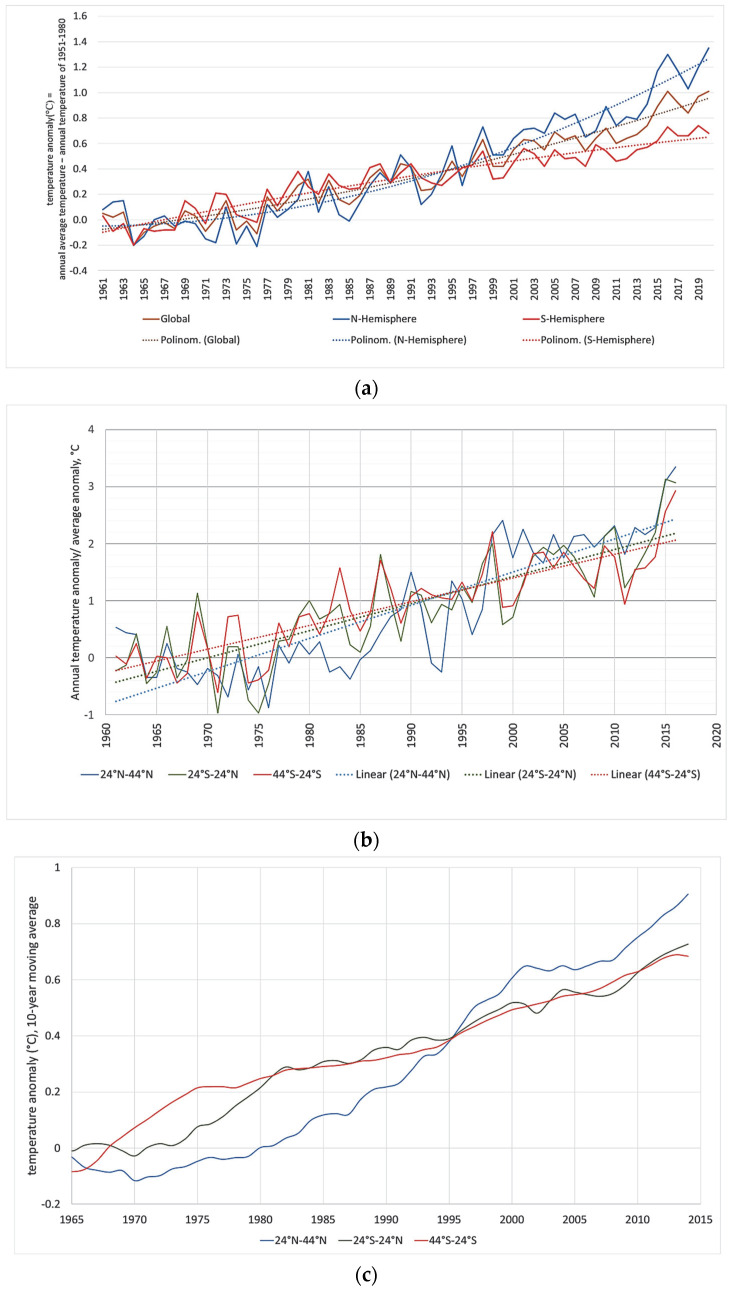
(**a**) Global temperature anomalies (annual mean temperature−mean temperature of 1951–1980). (**b**) Normalized annual temperature anomalies and their linear trends for the coffee-producing latitudes. (**c**) Ten-year moving averages of temperature anomalies for the coffee-producing latitudes. Source: authors’ own construction based on data from [51].

**Figure 3 foods-11-03036-f003:**
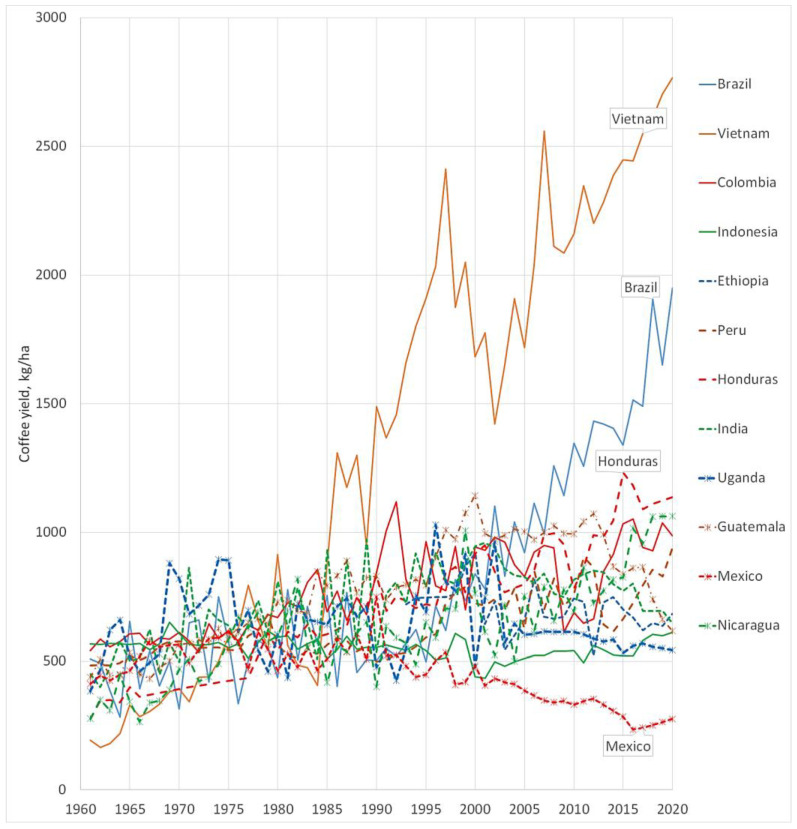
Coffee (green beans) yield tendencies from 1961 to 2020 (kg/ha). Source: authors’ own construction based on data from [50].

**Figure 4 foods-11-03036-f004:**
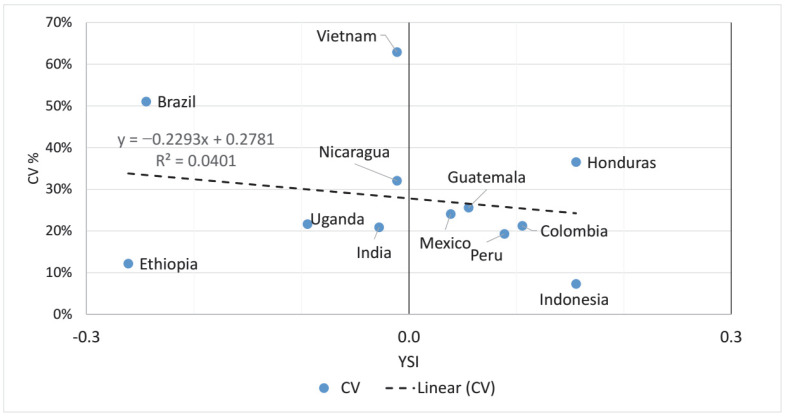
YSI vs. CV% scatterplot for coffee yields, 1961–2020, 12 countries. Source: authors’ own elaboration.

**Figure 5 foods-11-03036-f005:**
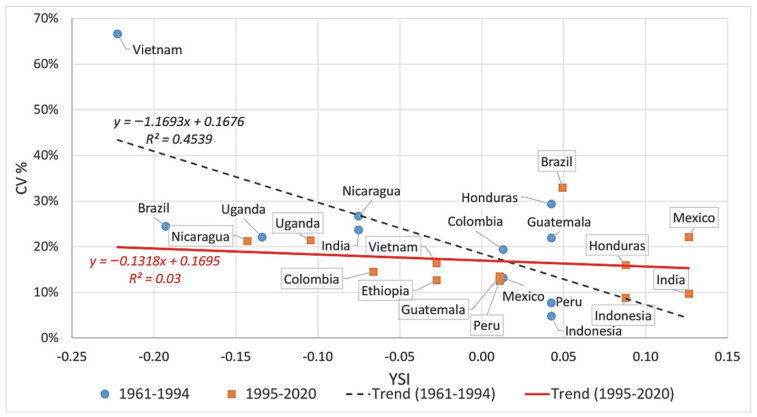
YSI vs. CV% scatterplot for coffee yields, 1961–1994 and 1995–2020. Source: authors’ own elaboration.

**Figure 6 foods-11-03036-f006:**
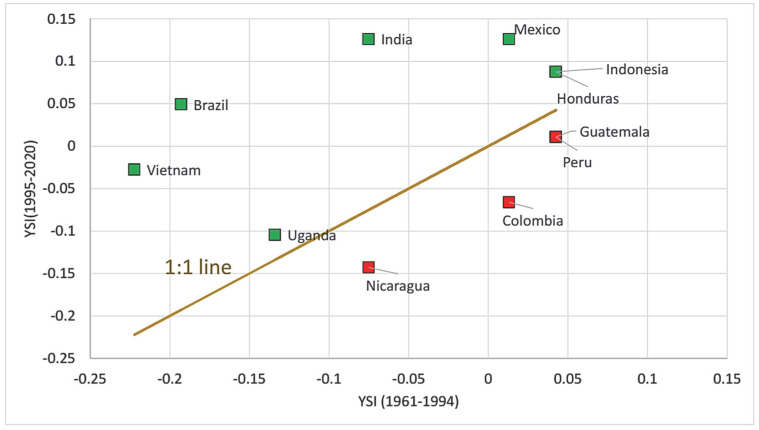
Comparison of Yield Stability Index values in 1961–1994 and 1995–2020. Source: authors’ own construction.

**Figure 7 foods-11-03036-f007:**
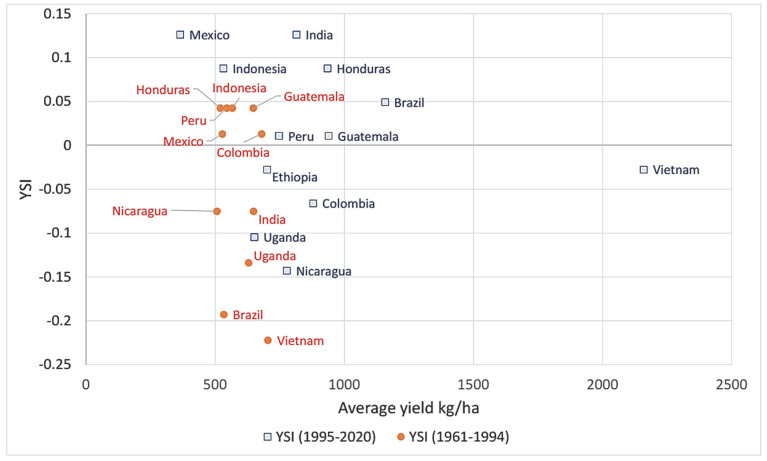
Comparison of YSI and average yield in 1961–1994 and 1995–2020. Source: authors’ own construction.

**Table 1 foods-11-03036-t001:** The share of countries in total global production and export of coffee green beans.

Rank	Continent	Countries	2020	1995	2020	2020
(2020)			Production Quantity, Tons	% in Production Quantity		% in Export Quantity
1	S America	Brazil	3,700,231	16.81	34.25	30.5
2	Asia	Vietnam	1,763,476	3.94	16.33	15.8
3	S America	Colombia	833,400	14.85	7.72	8.9
4	Asia	Indonesia	773,409	8.27	7.16	4.8
5	Africa	Ethiopia	584,790	4.16	5.41	3.0
6	S America	Peru	376,725	1.75	3.49	2.7
7	N America	Honduras	364,552	2.39	3.37	4.7
8	Asia	India	298,000	3.25	2.76	2.6
9	Africa	Uganda	290,668	3.28	2.69	4.2
10	N America	Guatemala	225,000	3.81	2.08	2.4
11	N America	Mexico	175,555	5.87	1.63	1.3
12	N America	Nicaragua	158,759	0.99	1.47	1.9
TOTAL selected				69.38	88.36	83.1
Global total			10,802,153	100.00	100.00	100.00

Source: Authors’ own construction based on data from [24].

**Table 2 foods-11-03036-t002:** The share of coffee compared to national export and import values.

Countries	Coffee Share in Total Agricultural Export		Coffee Share in Total Merchandise Export		Coffee Export as % of Total Food Import	
	2018	2020	2018	2020	2018	2020
(1)	(2)	(3)	(4)	(5)	(6)	(7)
Brazil	5.3%	5.8%	1.8%	2.4%	54.9%	55.7%
Colombia	32.0%	32.0%	5.4%	7.9%	45.9%	48.6%
Ethiopia	32.4%	47.7%	13.3%	22.8%	17.0%	49.3%
Guatemala	12.9%	10.7%	6.3%	5.7%	27.9%	23.7%
Honduras	47.1%	37.8%	12.8%	12.8%	75.9%	61.3%
India	1.7%	1.4%	0.2%	0.2%	2.8%	2.5%
Indonesia	2.2%	2.2%	0.4%	0.5%	5.4%	5.8%
Mexico	1.1%	1.0%	0.1%	0.1%	1.6%	1.9%
Nicaragua	20.3%	19.8%	8.3%	8.6%	51.0%	51.8%
Peru	10.2%	8.5%	1.4%	1.5%	18.0%	16.0%
Uganda	27.0%	33.8%	14.1%	12.4%	58.0%	59.7%
Vietnam	15.0%	10.6%	1.2%	0.7%	22.2%	13.5%

Source: Authors’ own construction based on data from [24].

**Table 3 foods-11-03036-t003:** Prevalence of food insecurity and undernourishment (3-year average, % of the total population).

Population Affected by…	Severe Food Insecurity, %		Undernourishment, %	
%	2016–2018	2018–2020	2016–2018	2018–2020
(1)	(2)	(3)	(4)	(5)
Brazil	1.7	3.5	2.5	2.6
Colombia	na	na	5.9	7.2
Ethiopia	14.8	16.4	15.7	21.9
Guatemala	17.1	19.2	16.3	16.3
Honduras	14.1	14.6	13.2	13.3
India	na	na	13.2	14.6
Indonesia	0.9	0.7	5.9	6.2
Mexico	3.3	3.9	6.1	6
Nicaragua	na	na	17.2	17.5
Peru	16.6	19.2	7.6	8.1
Uganda	24.5	23.3	no data	no data
Vietnam	0.5	0.5	7.2	6.2

Source: Authors’ own construction based on data from [26].

**Table 4 foods-11-03036-t004:** Characteristics of the linear trends and 10-year moving averages for the coffee-producing latitudes.

	(1)	(2)	(3)	Max–Min of Trend Values	Absolute Difference in Trend Values			SUM of Absolute Diff’s	10-Year Moving Averages		
Year	24 N–44 N Trend	24 S–24 N Trend	44 S–24 S Trend		(1)–(2)	(2)–(3)	(1)–(3)		24 N–44 N	24 S–24 N	44 S–24 S
1990	0.921217	0.949045	0.982343	0.06113	0.0278	0.03330	0.061	0.122	0.218	0.359	0.322
1991	0.979352	0.996491	1.02391	0.04456	0.0171	0.02742	0.045	0.089	0.231	0.352	0.333
1992	1.037487	1.043937	1.065477	0.02799	0.0065	0.02154	0.028	0.056	0.277	0.385	0.338
1993	1.095622	1.091383	1.107044	0.01566	0.0042	0.01566	0.011	0.031	0.327	0.395	0.351
1994	1.153757	1.138829	1.148611	0.01493	0.0149	0.00978	0.005	0.030	0.335	0.385	0.36
1995	1.211892	1.186275	1.190178	0.02562	0.0256	0.00390	0.022	0.051	0.379	0.39	0.384
1996	1.270027	1.233721	1.231745	0.03828	0.0363	0.00198	0.038	0.077	0.441	0.42	0.412
1997	1.328162	1.281167	1.273312	0.05485	0.0470	0.00785	0.055	0.110	0.502	0.45	0.433
1998	1.386297	1.328613	1.314879	0.07142	0.0577	0.01373	0.071	0.143	0.528	0.475	0.455
1999	1.444432	1.376059	1.356446	0.08799	0.0684	0.01961	0.088	0.176	0.551	0.495	0.474

Source: authors’ own computation based on data from [51].

**Table 5 foods-11-03036-t005:** Average coffee yields (kg/ha), CV, and YSI for 1961–2020, with normality testing of residual series.

Countries	Average	CV%	YSI 1961–2020	Weakly Technologized	Shapiro–Wilk Test
(1)	(2)	(3)	(4)	(5)	*p* > 0.05 (0.01)
Brazil	804.0	51.1%	−0.245	x	0.1528 *
Vietnam	1334.4	62.9%	−0.011	x	0.3406 *
Colombia	766.1	21.2%	0.105		0.1861 *
Indonesia	551.2	7.3%	0.155		0.0495 **
Ethiopia	703.0	12.2%	−0.261	x	0.0439 **
Peru	632.3	19.3%	0.089		0.6117 *
Honduras	699.5	36.6%	0.155		0.8438 *
India	720.1	20.9%	−0.028	x	0.2597 *
Uganda	638.7	21.6%	−0.095	x	0.3678 *
Guatemala	773.6	25.6%	0.055		0.8130 *
Mexico	456.6	24.1%	0.039		0.8873 *
Nicaragua	624.1	32.1%	−0.011	x	0.3720 *

x: denotes weakly technologized countries; *: normality of residuals accepted at 0.05; **: normality of residuals accepted at 0.01. Source: authors’ own elaboration.

**Table 6 foods-11-03036-t006:** Average yields and yield fluctuations for 1961–1994 and 1995–2020.

Countries	1961–1994			1995–2020			YSI(2)–YSI(1)
	AVG Yield	CV%	YSI(1)	AVG Yield	CV%	YSI(2)	
(1)	(2)	(3)	(4)	(5)	(6)	(7)	(8)
Brazil	533.4	24.5%	−0.1929	1157.9	32.9%	0.0493	0.2422
Vietnam	704.0	66.6%	−0.2223	2158.8	16.4%	−0.0277	0.1946
Colombia	679.7	19.4%	0.0130	879.0	14.5%	−0.0661	−0.0791
Indonesia	566.1	4.8%	0.0424	531.6	8.8%	0.0877	0.0453
Ethiopia	732.7	2.4%	n.a.	700.7	12.6%	−0.0277	−0.0113
Peru	544.6	7.7%	0.0424	747.0	12.5%	0.0108	−0.0316
Honduras	519.8	29.3%	0.0424	934.6	15.9%	0.0877	0.0453
India	648.3	23.6%	−0.0752	814.1	9.7%	0.1262	0.2014
Uganda	629.1	22.1%	−0.1340	651.3	21.4%	−0.1046	0.0294
Guatemala	647.3	21.9%	0.0424	938.7	13.5%	0.0108	−0.0316
Mexico	527.4	13.1%	0.0130	363.9	22.1%	0.1262	0.1132
Nicaragua	506.8	26.7%	−0.0752	777.5	21.2%	−0.1430	−0.0678

n.a.: data not available. Source: authors’ own computation.

## Data Availability

Data for the present analysis can be downloaded from the publicly accessible FAOSTAT database using the following links: https://www.fao.org/faostat/en/#data/FS (food security indicators, accessed on 1 June 2022); https://www.fao.org/faostat/en/#data/TCL (coffee trade indicators, accessed on 1 June 2022); http://www.fao.org/faostat/en/#data/QC (coffee yield data, accessed on 1 June 2022); and http://data.giss.nasa.gov/gistemp/ (GISS Surface Temperature Analysis (GISTEMP dataset), accessed on 1 September 2021).

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
