# Peer review of "Coffee Yield Stability as a Factor of Food Security"

_foods, 2022, doi:10.3390/foods11193036_

Round 1
Reviewer 1 Report
This manuscript entitled “Coffee yield stability as a factor of food security” explores the stability of coffee yield with climate change and identity the vulnerable areas of the world. These contents are interesting and the results are helpful in making policies, although this manuscript looks simple more or less. However, a major revision is required for the several issues below.
(1) The background is too long in the first section, please concise it.
(2) The trends of the global temperature, as well as in both northern and southern hemispheres, are fitted by the quadratic polynomial, is there any problem? The quadratic and higher order polynomials can lead to a much larger trend ore increase rate with the increase of time.
(3) Two periods, before and after 1995, are not meaning based on the intersection of the trend lines, because the break point probably varies from each country for the spatial heterogeneity of climatic factors. Please test the break point for each country using a specific statistical method, and make the final conclusion.
(4) Are all the residuals of these data follow normal distribution? if not, the YSI means nothing. Please add the hypothesis testing.
(5) The figures are rough, please reproduce them. Some labels are them in a figure, e.g, both are YIS (1995-2020) in Figure 6, please confirm it and check the others.
(6) Is there any correlation between these indices and the climatic factors? If possible, please try to analyze it.
Author Response
Reviewer 1
The authors are grateful for the two anonymous reviewer for the valuable comments and recommendations for improving the paper. In line with the reviewer's recommendations the following changes have been made:
(1) The background is too long in the first section, please concise it.
The background was shortened, the Introduction section was restructured, repetitions and irrelevant details deleted.
(2) The trends of the global temperature, as well as in both northern and southern hemispheres, are fitted by the quadratic polynomial, is there any problem? The quadratic and higher order polynomials can lead to a much larger trend or increase rate with the increase of time.
See new text in page 8: The choice of the quadratic trend was motivated by the change patterns of the northern and southern hemispheres. The GISS temperature series in [51] show that the trends of global temperature anomalies follow a nonlinear trend, to which the lowest degree polynomial to fit the quadratic one. As Figure 2 shows, at the beginning of the analysed period the temperature change patterns are approximately flat in both hemispheres, while by the end the trends are quite steep – so a nonlinear trend line is more reasonable than a linear one. However, 0higher-order polinomials did not give a reasonably improved fit, and the coefficients for the higher-order factors were all very small (of the degree 10-6 or smaller). Thus the quadratic trend equations are used for the analysis. Simple algebra shows that the trend lines of the N-hemisphere and the S-hemisphere meet at year 1994 (x=34, year = 34+1960 = 1994).
Additional information, not included in the paper:
The quadratic equations are:
- Temp(Glob) = 0.000143x2 + 0.008748x - 0.08414, R2=0.9168
- Temp(N-hemisphere) = 0.000362x2 + 0.000216x - 0.04971, R2=0.9038
- Temp(S-hemisphere) = -0.0000703x2 + 0.01696x - 0.11442, R2=0.8739
Then the crosspoints are computed: Temp(N) = Temp(S):
Solution:
0.000362x2 + 0.000216x - 0.04971= -0.0000703x2 + 0.01696x - 0.11442
(0.000362+0.0000703) x2+ (0.000216 - 0.01696)x - 0.04971+0.11442 =0
0.0004323x2 - 0.016744x +0.06471 =0; from here:
x(1,2) = [0.016744(+-) sqrt(0.000280362-0.000111897)] / 0.0008646 = [ 0.016744± sqrt(0.000168465)] / 0.0008646 = [0.016744 ± 0.012979407] / 0.0008646 = 19.366 ± 15.012 = 4. 354 or 34.38
Using the notation 1961 = year 1, the first crosspoint is 1961+4 = 1965, the second one is 1961+34 = 1995.
These are visible in Fig 2 of the original submitted manuscript, too
(3) Two periods, before and after 1995, are not meaning based on the intersection of the trend lines, because the break point probably varies from each country for the spatial heterogeneity of climatic factors. Please test the break point for each country using a specific statistical method, and make the final conclusion.
Modified text from bottom of p8 to top of p10, and Fig 2b) and Fig2c), and a new Table 4 are added:
Countrywise temperature anomalies time-series were not available for the present research – but there are time series for the various latitude ranges relevant for coffee producing regions, i.e. the 24°N-24°S tropical zone, and for the 44°N-24°N zone and the 44°S-24°S. The ’normalised’ temperature change data (i.e. values divided by the series 60 year-average), show increasing linear-like trends, with the tropical zone having the highest values at the beginning, but the lowest ones by the end of the analysed period, with the smallest linear slope, while the 44°N-24°N zone, starting from the lowest level and ending at the highest level, has the steepest slope. The relevant linear trend lines are:
44°N-24°N: y = 0.058135x-114.767433; 24°N-24°S: y = 0.041567x-81.735987;
44°S-24°S: y = 0.047446x-93.468495
Comparing the three trend lines, they meet approximately at 1994-1995. The 10 – year moving averages for the three relevant zones (24N-44N, 24S-24N, 24S-44S) also show a breakpoint at 1995: up to 1995 the highest moving average is the 24S-24N region, and the smallest one is the 24N-44N zone, but from year 1996 the highest value is 24N-44N(the former coolest region), while the lowest value 24S-44S,and the 24N-24S region is between them. The 1990-1999 section of the time series is illustrated in Table 4. This supports the idea that it is meaningful to use the year 1994-1995 as the break-point for our analysis.
Two new figures, and a new Table 4 are added:
Figure 3b) Normalised annual temperature changes and their linear trends for the coffee producing latitudes
Figure 2c) 10 year moving averages of temperature anomalies for the coffee-producing latitudes
Table 4: Characteristics of the linear trends and 10-year moving averages for the coffee producing latitudes.
(4) Are all the residuals of these data follow normal distribution? if not, the YSI means nothing. Please add the hypothesis testing
See modified text and modified Table 5 (formerly Table 4) in page 12:
The residual series were tested for normality by the Shapiro-Wilk test. The p-values of the test are shown in the last column of Table 5. As these values indicate, all residual series are of normal distribution at the 0.05 or 0.01 level, therefore the logic of the YSI computation comparing the residuals to a normal distribution, are meaningful.
(5) The figures are rough, please reproduce them. Some labels are them in a figure, e.g, both are YIS (1995-2020) in Figure 6, please confirm it and check the others.
Fig 6 is corrected, other figures are also corrected – and Ethiopia is deleted from figures presenting YSI for 1961 -1994, because its data series start only from 1991, so YSI for the earlier period is not reliable. All figures are provided in higher resolution in a separate ZIP file.
(6) Is there any correlation between these indices and the climatic factors? If possible, please try to analyze it.
The following explanation is added to p18, 2nd paragraph of conclusions:
The relationship between YSI and climate variables is an exciting issue. However, YSI is one value for a series of years, therefore we have only 3 values for each country: one for 60 years, one for 34 years and one for 26 years. Computing correlations with 3 values is not very meaningful. A future research topic can be to compare YSI values for shorter time segments – e.g. 10-year periods – and compared to the 10-year mean temperature changes to see how YSI values are influenced by changing temperatures.
Reviewer 2 Report
1. The abstract must contain an introduction, objective, materials and methods, results and conclusion.
2. The introduction is excessive, please limit it to a single section.
3. Although the manuscript makes clear the relationship between yield stability and climate change, it does not manage to specify the concept of food security.
4. By not making clear the relevance and importance of food safety, the article is outside the scope of the journal.
Author Response
REVIEWER 2
The authors are grateful for the anonymous reviewer for the valuable comments and recommendations for improving the paper. In line with the reviewer's recommendations the following changes have been made:
The abstract must contain an introduction, objective, materials and methods, results and conclusion.
The abstract has been better structured indicating the suggested sections. The new text is as follows (p1):
Introduction: Yield fluctuation is a major risk in all agricultural sectors, and influences Goal 2, food security, of UN SDGs. Yield fluctuations are expected due to climate change, risking stable coffee supplies, and compromising coffee exporter countries ability to earn revenues to pay for food imports. Technology minimizing yield fluctuations is crucial for food security and coffee farmers’ stable income. Fluctuations are small if yields remain close to mean yield trends. Objective: Coffee yields of major producers are analyzed, together with zonal temperature data, to see where coffee is grown with stable technology under rising temperatures, demonstrating the advantages of a Yield Stability Index (YSI) over traditional stability measurements, in guiding policy formulation and managerial decisions. Methodology: A yield stability index (YSI) is applied for 1961-1994 and 1995-2020, for the world’s 12 major coffee producing countries. Results: YSI indicates that of the 12 countries only Indonesia, Honduras and Mexico maintain stable yield levels, while Brazil and Vietnam considerably improved their yield stability, which traditional stability measures cannot grasp. Conclusion: Country-wise differences exist in environmental vulnerability and adaptability, with implications on food security. The novelty is the application YSI, and the connection between yield stability, climate change and food security.
- The introduction is excessive, please limit it to a single section.
p1-p8: The Introduction has been shortened considerably, and sub-titles removed. However, as Recommendation 3) and Recommendation 4) required further elaboration of the issues of Food security and Food safety, these sections increased somewhat the size of Introduction, so shortening of it has limitations..
- Although the manuscript makes clear the relationship between yield stability and climate change, it does not manage to specify the concept of food security.
The following paragraph has been added in p2- from the 3rd paragraph, to explain food security, together with 3 new references relevant to food security and food safety.
Food security refers to a situation when all people, at all times, have physical, social and economic access to sufficient, safe and nutritious food for an active and healthy life. This definition includes the concepts of food availability, food access, and how food is utilized. Small-scale coffee producers are trying to maintain a sustainable livelihood with modest land holdings, high levels of initial capital investments in their coffee plants and a vulnerability to a volatile international price structure for their cash crop, living in countries with relatively weak trade positions. With higher coffee prices in the international market, the desire for a profitable cash crop often encourages farmers in traditional coffee growing areas to increase their production, leaving less and less areas for food subsistence. Export crops, such as coffee, may offer the promise of a better life, and an escape from the poverty trap of subsistence agriculture. Wishing to participate in a cash economy, many smallholder coffee farmers allocate their investments toward coffee and away from subsistence food production, hoping that extra money will allow for additional food purchases. The isolated rural areas where the world’s best coffee is grown are exposed to multiple food insecurity risk factors: depletion of natural resources,environmental degradation, shocks such as natural disasters , seasonal changes in food production and food prices.. Coffee farmers face the instability of green bean coffee prices, and of fluctuations in food prices, and these increase the food vulnerability of these communities (Caswell et al., 2012).
- By not making clear the relevance and importance of food safety, the article is outside the scope of the journal.
The following paragraph has been added in p2- from the bottom of p2 to top of p4 to explain food safety and its relations to food security, together with 3 new references
. Food safety is linked to food security through health and livelihoods. The presence of food hazards can lead to food losses and reduced food availability for food insecure populations. Food safety also increasingly plays a role in producer livelihoods, as smallholders seek to meet requirements in high value markets, particularly exports. As consumers often lack information about food safety hazards in in specific food products, thus cannot reward producers for supplying safer food, therefore producers may not have sufficient financial resources for the extra costs of safer food. Sometimes producers may also have little idea of existing hazards in their products, so they cannot react to improve food safety. (Unnevehr, 2015). In developing countries, efforts to improve food safety have been focused particularly on exports to high income countries. Compliance with food safety standards in high-income countries demonstrates the costs of such improvements, and how compliance leads to higher incomes for developing country smallholders, thus supporting improved food security (Unnevehr, 2015). To satisfy demands for food safety, retailers and manufacturers increasingly use sustainability-oriented standards and labels, especially for luxury foods, such as coffee, tea, cocoa, or tropical fruits. For coffee, the global market share of products with sustainability certification has been growing since 2006 (ITC, 2011; Chiputwa et al., 2015). Certified coffees are assumed to have higher price, but in reality, effective price advantages are often minimal for some farmers. As data from coffee farmers in Mexico and Peru show, yields may be more important than price premiums for increasing net returns. There are considerable regional differences in average yield and quality levels, that influence the incomes and livelihoods of coffee farmers, influencing food security of coffee growing rural regions (Chiputwa, 2015).
New references added:
Caswell, M., V.E. Méndez & C.M. Bacon (2012).Food security and smallholder coffee production: current issues and future directions. ARLG Policy Brief # 1. Agroecology and Rural Livelihoods Group (ARLG), University of Vermont. Burlington, VT, USA. Available online: http://www.uvm.edu/~agroecol/?Page=Publications.html
Unnevehr, L. Food safety in developing countries: Moving beyond exports. Global Food Security 4 (2015) 24–29. http://dx.doi.org/10.1016/j.gfs.2014.12.001
Chiputwa, B.; Spielman, D. J.; Qaim, M. Food Standards, Certification, and Poverty among Coffee Farmers in Uganda. World Development Vol. 66, pp. 400–412, 2015. http://dx.doi.org/10.1016/j.worlddev.2014.09.006
Round 2
Reviewer 1 Report
On the basis of these detailed responses and revisions, I recommend to a minor revision, although there is still some small issues . e.g, the Y-axis of Figure 2 should be labeled by anomaly that equals to the average annual temperature minus the mean, curves in Figure 3 are not easy to see. Please modify them further and check the others.
Author Response
Comments and Suggestions for Authors by Reviewer 1: On the basis of these detailed responses and revisions, I recommend to a minor revision, although there is still some small issues . e.g, the Y-axis of Figure 2 should be labeled by anomaly that equals to the average annual temperature minus the mean, curves in Figure 3 are not easy to see. Please modify them further and check the others.
REPLY to REVIEWER1:
The authors gratefully thank the reviewer for the careful assessment of our submission and for pointing out further needs of improvements.
The suggested improvements of Fig 2a-2b-2c were carried out, Y-axis titles added and modification of labels and Figure captions was done.
In Figure 3 the colours and line types of the curves were adjusted to make them distinguishable, and the size of the figure was also enlarged for better visibility.
The other figures were also checked for axis labels and other notations.
Reviewer 2 Report
1. Abstract. Please remove the words introduction, objective, methodology, results and conclusion.
2. Although you have tried to summarize the introduction, it should be more concrete.
Author Response
Comments and Suggestions for Authors by Reviewer 2:
- Abstract. Please remove the words introduction, objective, methodology, results and conclusion.
- Although you have tried to summarize the introduction, it should be more concrete.
REPLY to REVIEWER 2:
The authors gratefully thank the reviewer for the careful assessment of our submission and for pointing out further needs of improvements.
The Abstract was modified, the words „introduction, objective, methodology, results and conclusion” were deleted, as suggested by the reviewer.
The Introduction was further shortened, the too general parts deleted and the the focus on more concrete details was enhanced. Altogether the Introduction has been shortened by more than one full page, especially the parts about the climate change effects, the coffe production parts and the general aspects of yield fluctuation measurements.